# Coevolution of *furA*-Regulated Hyper-Inflammation and Mycobacterial Resistance to Oxidative Killing through Adaptation to Hydrogen Peroxide

Xin Fan,[a] Bei Zhao,[b] Weishan Zhang,[a,b] Ning Li,[a] ORCID Kaixia Mi,[a,b] ORCID Beinan Wang[a,b]

aCAS Key Laboratory of Pathogen Microbiology and Immunology, Institute of Microbiology, Chinese Academy of Sciences, Beijing, China
bSavaid Medical School, University of Chinese Academy of Sciences, Beijing, China

Xin Fan and Bei Zhao contributed equally to this work. The order of authors is determined by contribution.

**ABSTRACT**    *Mycobacterium tuberculosis* (*Mtb*) is highly resistant to host oxidative killing. We hypothesized that the evolutionary adaptation of *M. smegmatis* to hydrogen peroxide ($H_2O_2$) would endow the nonpathogenic *Mycobacterium* persistent in a host. In the study, we screened a highly $H_2O_2$-resistant strain (mc²114) via evolutionary $H_2O_2$ adaptation *in vitro*. The MIC of mc²114 to $H_2O_2$ is 320 times that of wild-type mc²155. Mouse infection experiments showed that mc²114, similar to *Mtb*, was persistent in the lungs and caused high lethality in mice with restricted responses of NOX2, ROS, IFN-$\gamma$, decreased macrophage apoptosis, and overexpressed inflammatory cytokines in the lungs. Whole-genome sequencing analysis revealed that mc²114 harbored 29 single nucleotide polymorphisms in multiple genes; one of them was on the *furA* gene that caused FurA deficiency-mediated overexpression of KatG, a catalase-peroxidase to detoxify ROS. Complementation of mc²114 with a wild-type *furA* gene reversed lethality and hyper-inflammatory response in mice with rescued overexpression of KatG and inflammatory cytokines, whereas NOX2, ROS, IFN-$\gamma$, and macrophage apoptosis remained reduced. The results indicate that although FurA regulates KatG expression, it does not contribute significantly to the restriction of ROS response. Instead, FurA deficiency is responsible for the detrimental pulmonary inflammation that contributes to the severity of the infection, a previously nonrecognized function of FurA in mycobacterial pathogenesis. The study also indicates that mycobacterial resistance to oxidative burst results from complex mechanisms involving adaptive genetic changes in multiple genes.

**IMPORTANCE**    *Mycobacterium tuberculosis* (*Mtb*) causes human tuberculosis (TB), which has killed more people in human history than any other microorganism. However, the mechanisms underlying *Mtb* pathogenesis and related genes have not yet been fully elucidated, which impedes the development of effective strategies for containing and eradicating TB. In the study, we generated a mutant of *M. smegmatis* (mc²114) with multiple mutations by an adaptive evolutionary screen with $H_2O_2$. One of the mutations in *furA* caused a deficiency of FurA, which mediated severe inflammatory lung injury and higher lethality in mice by overexpression of inflammatory cytokines. Our results indicate that FurA-regulated pulmonary inflammation plays a critical role in mycobacterial pathogenesis in addition to the known downregulation of NOX2, ROS, IFN-$\gamma$ responses, and macrophage apoptosis. Further analysis of the mutations in mc²114 would identify more genes related to the increased pathogenicity and help in devising new strategies for containing and eradicating TB.

**KEYWORDS** FurA, Mycobacterial pathogenesis, evolutionary hydrogen peroxide adaptation, pulmonary inflammation, resistance to oxidative killing

Address correspondence to Beinan Wang, wangbn@im.ac.cn, or Kaixia Mi, mik@im.ac.cn.

The authors declare no conflict of interest.

Tuberculosis (TB) caused by *Mycobacterium tuberculosis* (*Mtb*) is an important infectious disease associated with significant morbidity and mortality globally. Vaccine inefficiency, diagnostic challenges, and antibiotic resistance have hindered efforts at controlling TB. Understanding the mechanisms underlying the interaction between *Mtb* and the host will facilitate the development of targeted strategies against TB. Reactive oxygen species (ROS) are crucial for macrophage clearance of infected bacilli (1). *Mtb* is highly resistant to oxidative intermediates, which is recognized as the most notable characteristic for *Mtb* survival within host cells. However, the mechanisms underlying the antioxidant activities of *Mtb* are not understood. *Mycobacterium smegmatis* (*Msmeg*) is nonpathogenic *mycobacterium*, a model of *Mtb*. It is strikingly different from *Mtb*. Many genes in *Mtb* encode resistance to oxidative intermediators. These genes are also in *Msmeg*. Although *Mtb* has a diminished capacity to restore endogenous redox balance in comparison with *Msmeg* (2) it tolerates exogenous ROS and can persist in phagocytes. Whereas *Msmeg* is cleared from the lungs promptly, suggesting that *Msmeg* is less resistant to exogenous ROS (3), this is probably because activities of these genes in *Msmeg* differ markedly. We hypothesized that the evolutionary adaptation of *Msmeg* to $H_2O_2$ would induce changes in these genes in response to stress and make it more similar to *Mtb*. A previous study reported that an $H_2O_2$-resistant *Msmeg* strain (mc$^2$51) that emerged through evolutionary adaptation carries genetic alterations (4) and facilitates its survival in a host (3). However, the interactions between $H_2O_2$-adapted *Msmeg* and hosts and their underlying mechanisms and their association with genetic changes have not been established.

Ferric uptake regulators (FURs) are a superfamily of prokaryotic transcriptional regulators for the global regulation of various gene expressions in several bacteria (5). FUR paralogs control the defense against oxidative stress and metal homeostasis, which is associated with bacterial virulence (6). A FUR-like gene, *furA*, is found in several mycobacterial species, including *Mtb* and *Msmeg*. FurA in mycobacteria negatively regulates KatG, a catalase-peroxidase, to detoxify reactive oxygen species (7–9). However, the role of FurA in mycobacterial pathogenicity has not been fully explored.

In this study, we selected an $H_2O_2$-resistant strain of *Msmeg* that emerged through experimental evolutionary adaptation to $H_2O_2$. The $H_2O_2$-resistant strain (mc$^2$114) was genetically and phenotypically assessed. We found that mc$^2$114 carries multiple mutations in different genes and was persistent in mice with hyper-inflammation in the lungs and attenuated ROS-mediated bacterial elimination. Genetic complementation analysis indicates that *furA*, one of the genes that changed in mc$^2$114, is involved in the hyper-inflammatory response but not ROS attenuation in mice.

## RESULTS

**Evolutionary adaptation to $H_2O_2$ and the $H_2O_2$-resistant strain of *Msmeg* (mc$^2$114).** The wild-type *Msmeg* strain mc$^2$155 was cultured with increasing concentrations of $H_2O_2$, beginning with 0.0293 mM and continuously subculturing until 3.334 mM (Fig. 1A). mc$^2$114 was selected for phenotype analysis. The MIC of $H_2O_2$ for mc$^2$114 was 12.5 mM, which was 320-fold that of mc$^2$155 (0.039 mM), and the growth kinetics of mc$^2$155 and mc$^2$144 cultured in 7H9 showed that mc$^2$114 grows 38.9% slower than mc$^2$155 (Fig. 1B). Compared with mc$^2$155, colonies of mc$^2$114 appeared more humid and raised and had denser crenelated rims (Fig. 1C). Whole-genome sequencing revealed 29 single nucleotide polymorphisms (SNPs) in mc$^2$114, and most of them occurred in genes involved in metabolism (see Table S1 in the supplemental material). One of the SNPs was found in the *furA* gene; it was a single amino acid substitution of valine for alanine at position 28 (A28V) and the same SNP previously reported in the $H_2O_2$-resistant mutant mc$^2$51 (10). Western blotting showed that KatG expression and its enzymatic activity in mc$^2$114 were increased approximately 5-fold relative to those in mc$^2$155 (Fig. 1D), indicating that *furA* mutation results in KatG upregulation. Isoniazid (INH) is a frontline antimycobacterial agent (9, 11), and its activity is enhanced by KatG. We found that the MIC of INH for mc$^2$114 was dramatically reduced by approximately 50-fold that for mc$^2$155 (0.195 $\mu$g/mL for mc$^2$114 and 10 $\mu$g/mL for mc$^2$155). Taken together, these results indicate that multiple SNPs, including one in *furA* with changed features of *Msmeg*, emerged in adaptation to $H_2O_2$.

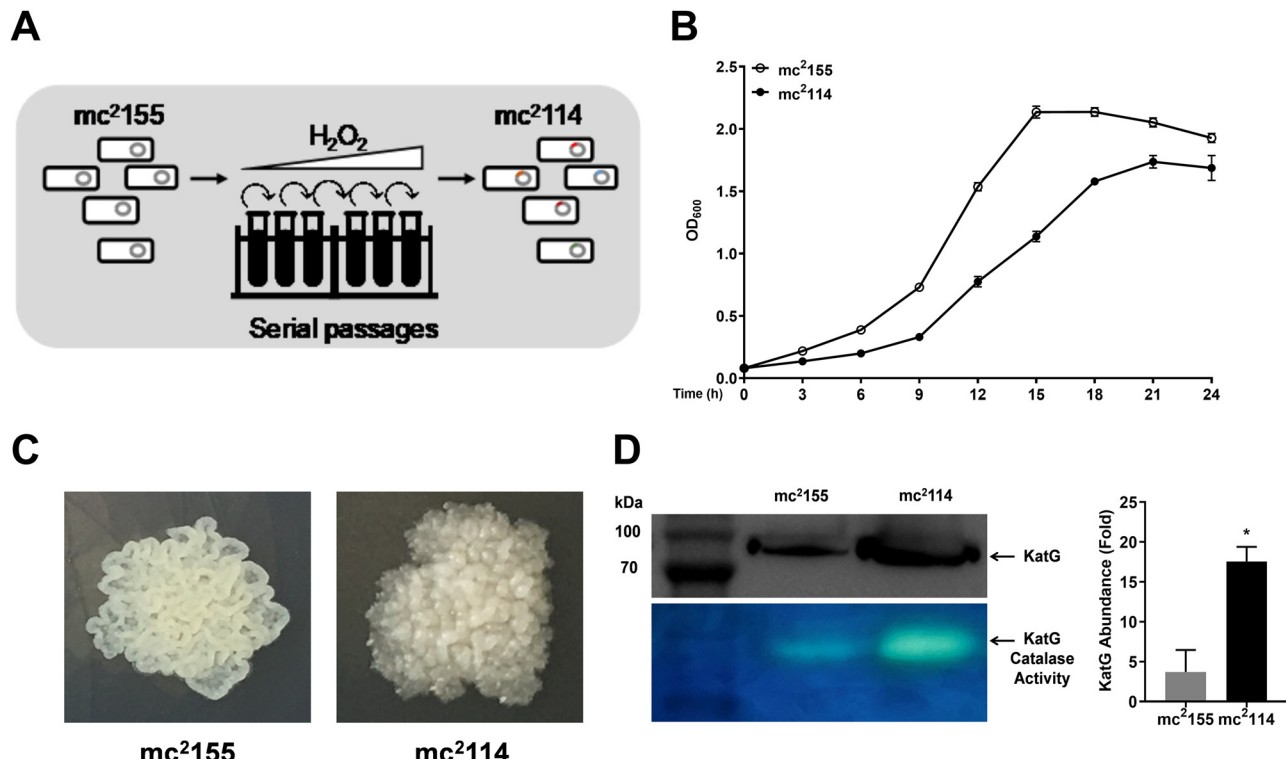

**FIG 1** Evolutionary selection and characterization of a high $H_2O_2$-resistant strain of *Msmeg*. (A) Overview of the experimental setup. The *Msmeg* strain mc²155 grew for 25 generations under increasing concentration of $H_2O_2$ in the medium. The $H_2O_2$-resistant mc²114 was selected from the final generation. (B) Growth curves of mc²155 and mc²114. (C) Colonial morphology of mc²155 and mc²114. (D) The expression of KatG (top) was measured by Western blotting and presented by densitometry quantification (bottom right). Enzymatic activity of KatG of mc²155 and mc²114 (bottom) was measured by the standard assay and analyzed in native polyacrylamide gels. Data are presented as means ± SEM from three independent experiments and representative images are shown. Asterisks indicate significant differences (*, $P < 0.05$ by two-tailed unpaired *t* tests).

**Increased intracellular survival and high lethality in mc²114-infected mice.** We hypothesized that the high resistance to $H_2O_2$ would facilitate mc²114 persistence in hosts. To test the hypothesis, mice were intranasally infected with $2.44 \times 10^7$ CFU of mc²155 or $1.51 \times 10^7$ CFU of mc²114/mouse, and bacterial loads in the lungs were determined. More CFU were detected in the single cell suspensions of lung tissue of mc²114-infected mice than mc²155-infected mice (Fig. 2A). Alveolar macrophages (AMΦs) were isolated from the bronchoalveolar lavage fluid (BALF) of mice to determine intracellular survived bacteria. We found that more intracellular bacteria were released from AMΦs of mc²114-infected mice compared with mc²155-infected mice (Fig. 2B). To rule out the possibility that the increase in AMΦs resulted in an increase in the number of CFU of mc²114, equal numbers of THP-1 cells (a human leukemia monocytic cell line, which has been extensively used to study monocyte/macrophage functions) were inoculated with mc²114 or mc²155 for intracellular survival determination. The intracellular CFU of mc²114 were 10-fold more than those of mc²155 in the THP-1 cells at all time points after inoculation. By 120 h postinfection (pi), no CFU were released from mc²155-inoculated THP-1 cells; approximately 100 CFU were recovered from mc²114-inoculated cells (Fig. 2C). Viability assay for THP-1 cells (CCK-8) showed that proliferation of THP-1 was significantly higher in the mc²155-infected than in mc²114-infected group at all examined time points (data not shown), ruling out that higher numbers of intracellular mc²114 is due to increased THP-1 cells. These results indicate that mc²114 survives better in AMΦs than mc²155. The mice were also inoculated intravenously ($3 \times 10^7$ CFU of mc²155 or mc²114) to determine its systemic virulence when the bacteria enter the circulation. We found that all the mc²114-infected mice died by day 5 pi, while 100% of mc²155-infected mice survived at the end of the experiment (Fig. 2D). These results indicate that mc²114 is better adapted to the host and more virulent than its parental strain mc²155.

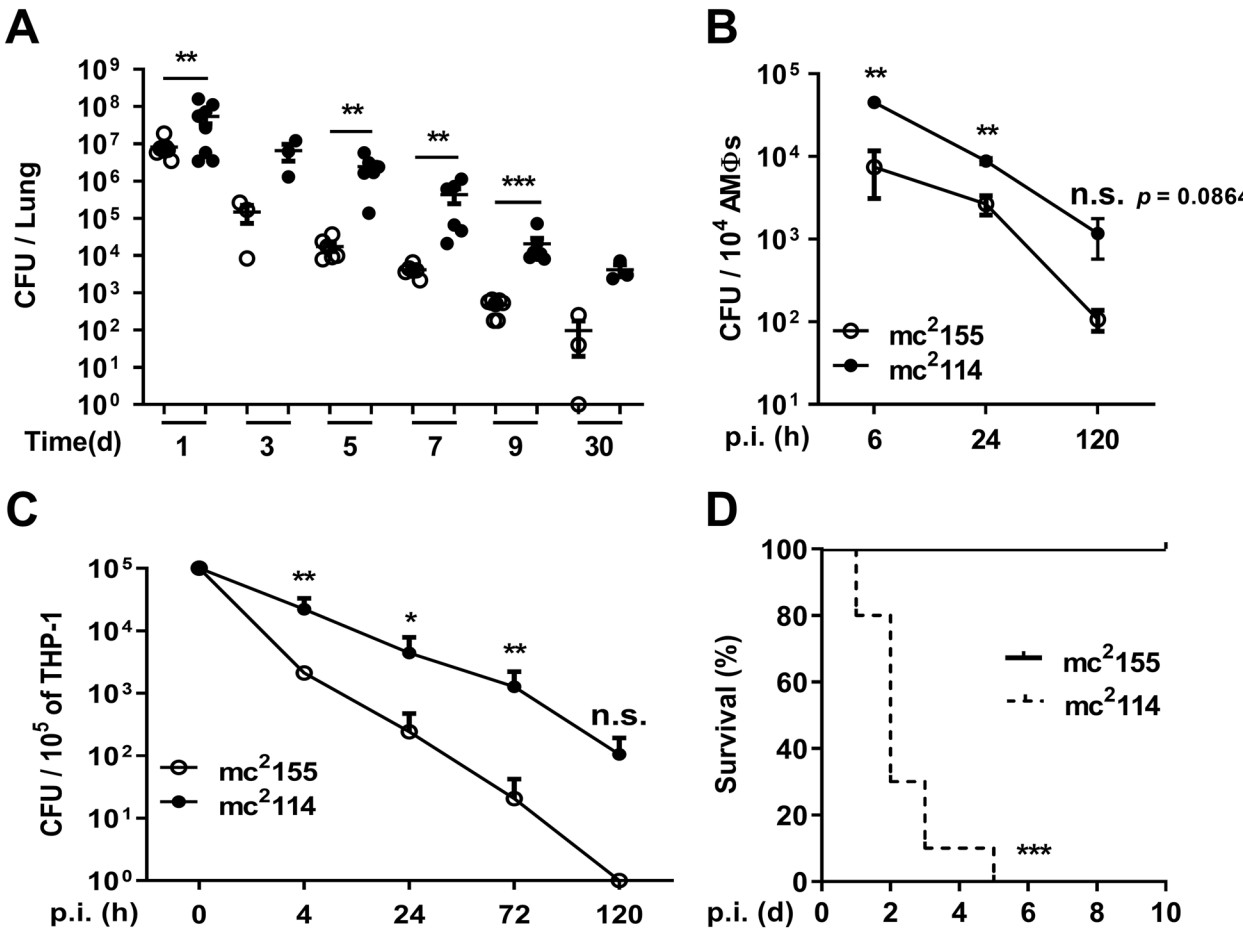

**FIG 2** $H_2O_2$-resistant mutant mc$^2$114 displayed increased lung colonization, intracellular survival, and lethality in mice. Mice were infected with $2.44 \times 10^7$ CFU of mc$^2$155 or $1.51 \times 10^7$ CFU of mc$^2$114/mouse on day 0 and sacrificed at indicated time points. CFU in the lungs (A) and the intracellular CFU in AMΦs (B) isolated from the BALF of mice were determined at different time points after mice were infected with mc$^2$155 or mc$^2$114. (C) THP-1 cells were infected and intracellular bacteria were determined. Data are presented as means $\pm$ SEM from two or three independent experiments. *, $P < 0.05$; **, $P < 0.01$; ***, $P < 0.001$, and n.s., not significant by two-tailed unpaired Mann-Whitney *t* tests. (D) Mice were intravenously (i.v.) infected with $3 \times 10^7$ CFU of mc$^2$155 or mc$^2$114 ($n = 10$) and the survival rate was monitored daily.

**mc$^2$114 decreased the abundance of ROS and dysregulated ROS-related antibacterial mechanisms.** Infection induces antimicrobial oxidative stress resulting in the accumulation of intracellular ROS (12, 13), and mycobacteria need to escape from ROS for intracellular survival (14). The intracellular persistence of mc$^2$114 suggests that it reduces ROS. The oxidative burst in AMΦs was determined by flow cytometry for the mean fluorescence intensity of dichlorofluorescein, which reflects the intracellular concentration of ROS. Compared with naive mice, the concentrations of ROS were higher in mc$^2$155-infected mice but the same in mc$^2$114-infected mice (Fig. 3A), suggesting that ROS response to mc$^2$114 is restricted.

KatG detoxifies $H_2O_2$ in culture (8), but reports on the role of KatG in hosts are conflicting (15, 16). The restricted ROS response may result from the increase in KatG in mc$^2$114 or diminished ROS production. The major ROS-producing enzymes are NADPH oxidases (NOXs) (17). Therefore, NOX2, the isoform of the NOX family found in professional phagocytes (18–20), was detected by quantitative real-time PCR (qRT-PCR) assays. NOX2 expression was strongly induced in the lungs of mc$^2$155- but not mc$^2$114-infected mice (Fig. 3B). The results of NADPH measurement showed that NADP$^+$/NADPH ratios were similar in mc$^2$114- and mc$^2$155-infected groups, though higher levels of NADPH and NADP$^+$ were found in the mc$^2$114-infected group than in the mc$^2$155-infected group (Fig. S1). These results indicate that NOX2 expression, but not NADPH levels, was restricted in mc$^2$114-infected mice and responsible for ROS reduction (21, 22).

IFN-$\gamma$ is crucial for ROS induction by triggering NOX2 expression in MΦs (23, 24). The inhibition of IFN-$\gamma$ production by *Mtb* leads to host susceptibility to *Mtb* (25). The NOX2

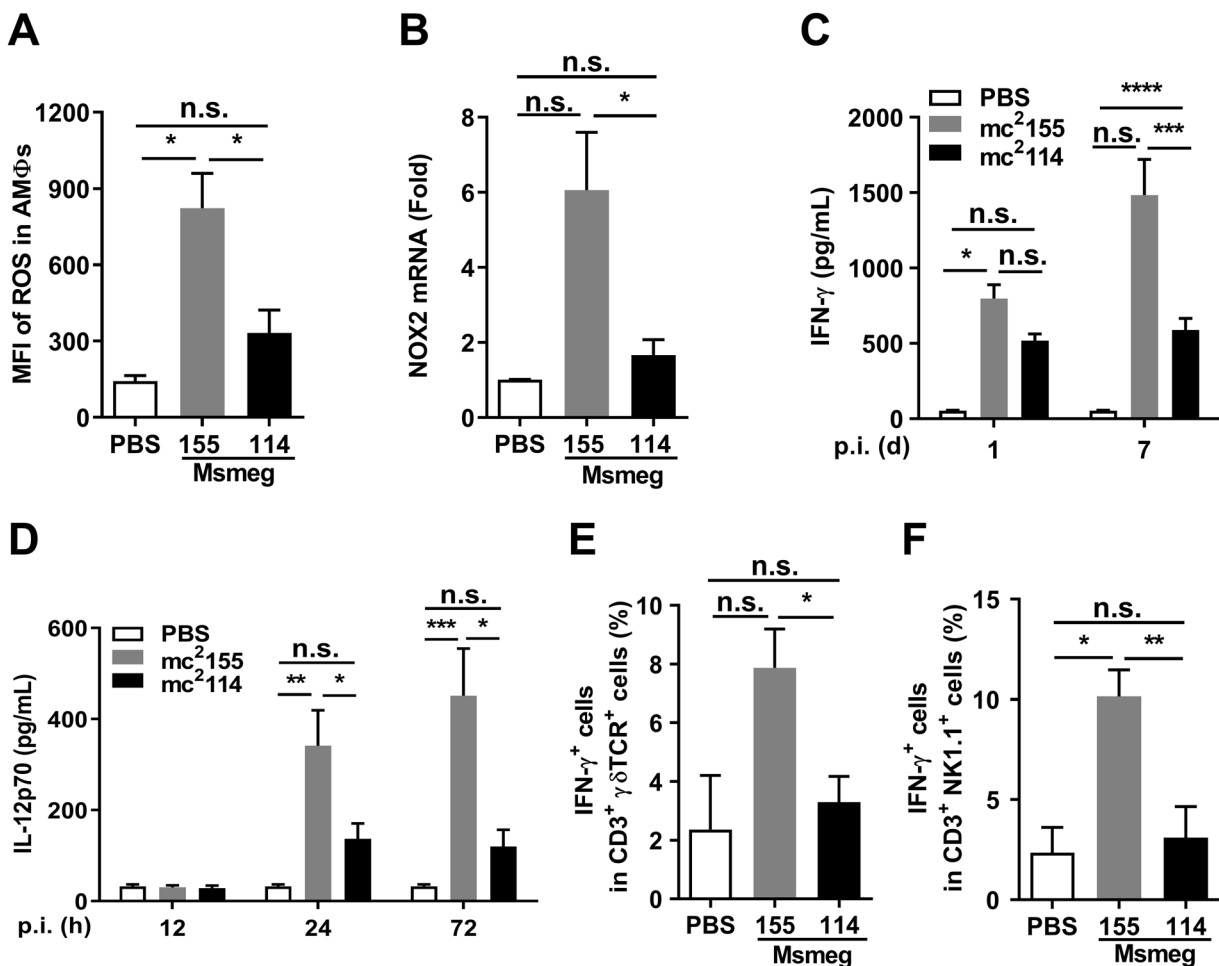

**FIG 3** mc²114 restricted ROS response and dysregulated ROS-related pathways. Mice were intranasally (i.n.) infected with mc²155 or mc²114. (A) ROS levels in AMΦ isolated from the BALF of mice were measured by flow cytometry 24 h after infection for means of fluorescence intensity (MFI). (B) NOX2 mRNA in the lung cells were analyzed by qRT-PCR. (C) The concentration of IFN-γ and (D) IL-12p70 in lung homogenate samples of infected mice were measured by ELISA. (E) Lung cells were analyzed by flow cytometry for frequencies of IFN-γ-secreting γδ T cells (CD3⁺ γδ TCR⁺ IFN-γ⁺) and (F) NKT cells (CD3⁺ NK1.1⁺ IFN-γ⁺) at 6 h after infection. Data are presented as means ± SEM from two independent experiments. (*, $P < 0.05$; **, $P < 0.01$; ***, $P < 0.001$; ****, $P < 0.0001$; and n.s., not significant by one-way ANOVA with Tukey's test).

reduction and better mc²114 survival in AMΦs suggest an impeded IFN-γ response to mc²114. Enzyme-linked immunosorbent assays (ELISAs) for the supernatants of lung homogenates revealed that the IFN-γ response was lower in mc²114-infected than in mc²155-infected mice (Fig. 3C). IL-12 is a critical cytokine to enhance IFN-γ induction (26). Consistently, the IL-12p70 in the lungs of mc²114-infected mice was markedly reduced relative to that in mc²155-infected mice (Fig. 3D). The γδ T and natural killer T (NKT) cells are the primary sources of innate IFN-γ (27, 28). Therefore, these cells were examined by flow cytometry for IFN-γ expression. We found that the IFN-γ⁺ γδ T and IFN-γ⁺ natural killer T cells in the lungs were substantially lower in the mc²114-infected than in the mc²155-infected mice (Fig. 3E and F). Taken together, the NOX2 reduction associated with restricted IFN-γ induction suggests that mc²114 constrains host NOX2 response through the inhibition of the IL-12-IFN-γ pathway.

MΦ apoptosis is a host mechanism for clearing intracellular mycobacteria, and it was impeded by ROS insufficiency (1, 29, 30). Flow cytometry analysis showed that fewer apoptotic cells were found in mc²114- than in mc²155-infected THP-1 cells (Fig. 4A). Mouse infection experiments confirmed that apoptotic lung cells were increased in mc²155-infected mice but not those infected by mc²114 6 h after infection (Fig. 4B). ROS promotes apoptosis by decreasing the expression of the antiapoptotic protein Bcl-2 (31, 32) and increasing Bax expression (33). Further analysis of THP-1 cells by Western blotting

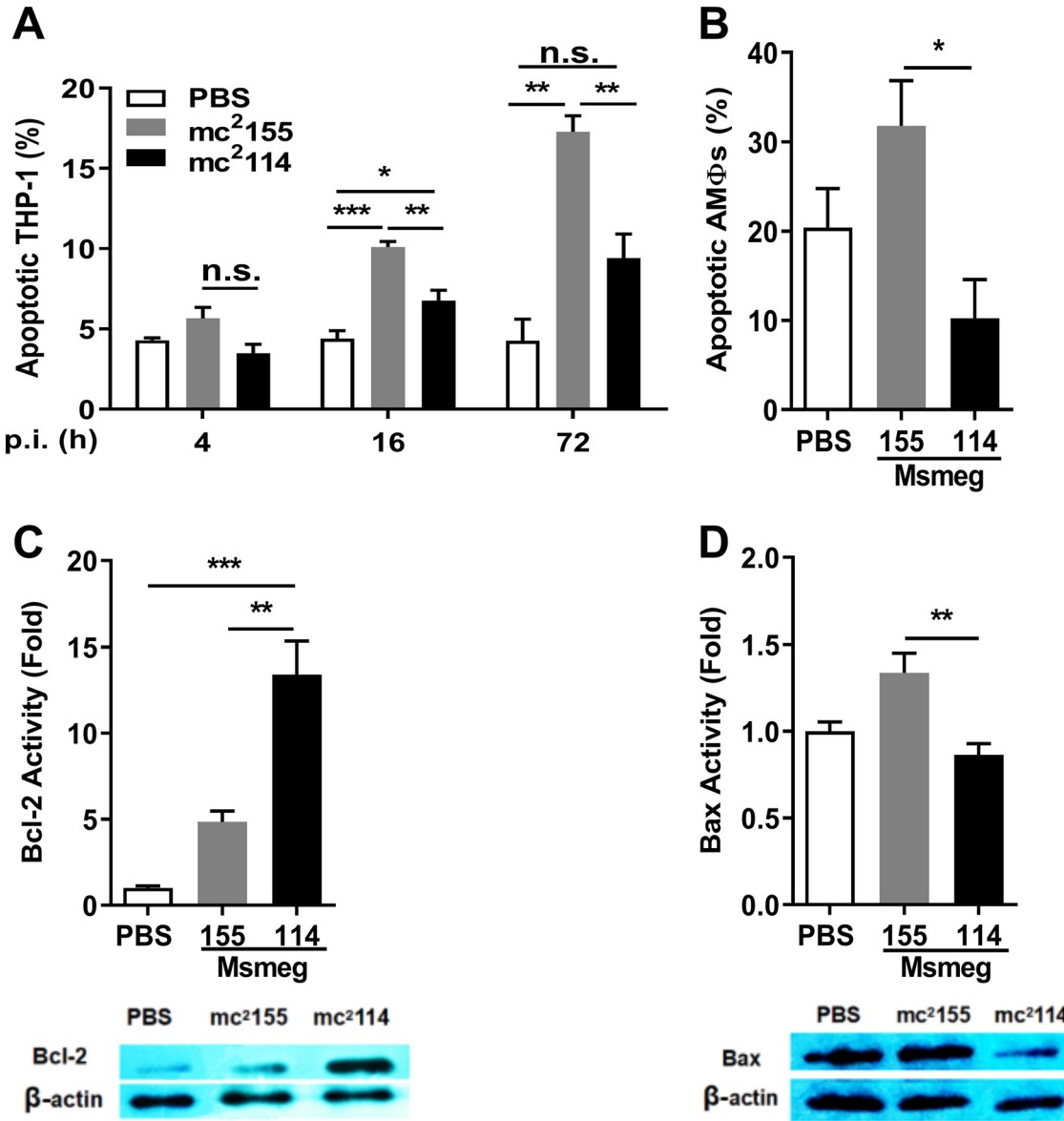

**FIG 4** MΦ apoptosis was inhibited in mc²114 infection. (A) THP-1 cells were infected with mc²155 or mc²114 at an MOI of 10. Uninfected THP-1 cells were inoculated with PBS. Samples were collected at indicated time points and stained apoptotic cells were determined by flow cytometry. (B) AMΦs were isolated from mice 6 h pi. and apoptotic cells were determined as in panel A. Data are presented as means ± SEM from two independent experiments. (C and D) THP-1 cells were infected with mc²155 or mc²114. Twelve hours after infection Bcl-2 (C) and Bax (D) were analyzed by Western blotting. Bcl-2 or Bax was quantified by densitometry and presented as amount of Bcl-2 or Bax relative to β-actin. Data are from a representative of two independent experiments and representative images are shown. (*, $P < 0.05$; **, $P < 0.01$; ***, $P < 0.001$; and n.s., not significant by one-way ANOVA with Tukey's test).

showed that significantly higher levels of Bcl-2 and lower levels of Bax were found in mc²114-infected cells than in mc²155-infected cells (Fig. 4C and D). These results indicate that mc²114 disrupts host apoptotic responses through the reduction of ROS (29).

**mc²114 induced higher levels of inflammatory cytokines and exacerbated pulmonary pathology.** *Mtb* can cause an inflammatory cascade, leading to extensive pathogenic damage in the lungs (34). To determine if its increased lethality in mc²114-infected mice was related to high inflammatory responses, inflammatory mediators in the lung homogenates were determined. ELISAs revealed higher levels of TNF-$\alpha$, IL-1$\beta$, and IL-6 in mice infected with mc²114 than in those infected with mc²155 (Fig. 5A–C). Consequently, higher numbers of neutrophils were observed in the lungs of mc²114-infected mice (Fig. 5D). In addition, lung sections revealed that mc²114 infection caused noticeable blood congestion and much less intact alveolar spaces (Fig. 5E) with heavier infiltration of inflammatory cells

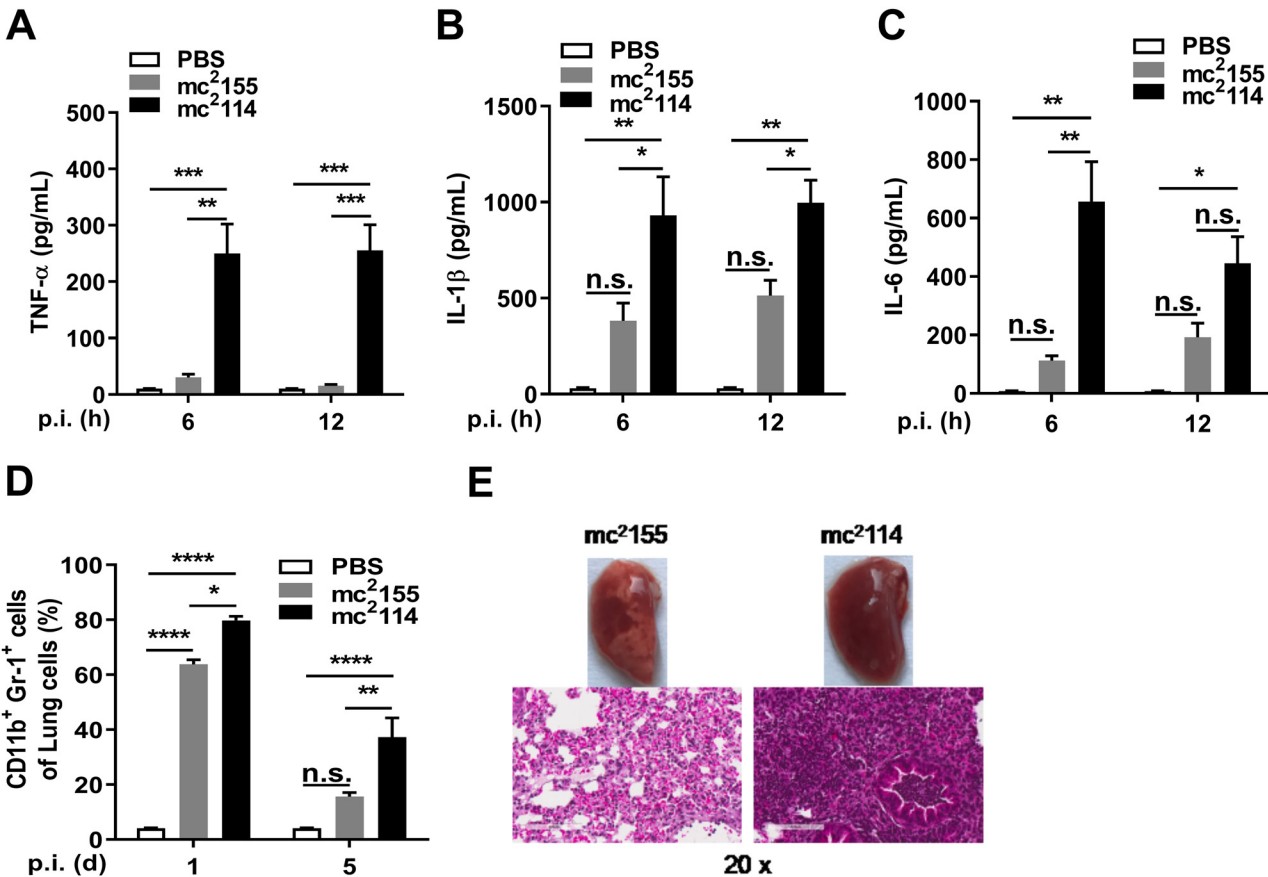

**FIG 5** mc²114 induced higher levels of inflammatory cytokines and exacerbated pulmonary pathology. The concentration of TNF-$\alpha$ (A), IL-1$\beta$ (B), and IL-6 (C) in lung tissue homogenates of mice were measured by ELISA at indicated time points after infection. (D) The neutrophils (CD11b⁺ Gr-1⁺) in the lungs were analyzed by flow cytometry at 24 h and 120 h pi. Data are means ± SEM from two or three independent experiments. (*, $P < 0.05$; **, $P < 0.01$; ***, $P < 0.001$; ****, $P < 0.0001$; and n.s., not significant by one-way ANOVA with Tukey's test). (E) Twenty-four hours after infection, lung sections were H&E stained (scale bar, 100 $\mu$m).

than in mc²155 infections. These observations indicate that mc²114 induces a severe pathological inflammatory response in the lungs.

***furA*-complemented mc²114 completely reversed the hyper-inflammatory but not ROS and ROS-related responses to mc²114.** The existence of the same SNP in the *furA* gene of mc²114 and the previously obtained H₂O₂-resistant mutant mc²51 suggests that FurA contributes to H₂O₂ resistance (3, 4). Electrophoretic mobility shift assays were used to assess the binding of the FurA protein with A28V mutation to target DNA (the promoter region of *furA*) and showed that DNA binding to the mutant *furA* decreased relative to that of the wild-type *furA* (see Fig. S2 in the supplemental material), indicating a mutant *furA* defect in mc²114. FurA is considered to contribute to *Mtb* persistence in host cells by regulating KatG and other genes (7). To determine the role of FurA in the pathogenicity of mc²114, a *furA*/mc²114 strain was generated by complementation of mc²114 with the wild-type *furA* gene (amplified from mc²155) (Fig. S3, Table S2). The generated *furA*/mc²114 grew slower than mc²155 and faster than mc²114 (Fig. 6A, top), and its colony morphology was closer to that of mc²114 than that of mc²155 (Fig. 6A, bottom). As expected, KatG expression and activity were significantly lower in *furA*/mc²114 than in mc²114 (Fig. 6B). These changes indicate that the complemented *furA* gene is functional in *furA*/mc²114.

THP-1 and mouse infection assays showed that CFU in *furA*/mc²114-infected THP-1 cells reduced to the levels found in mc²155-infected cells (Fig. 6C). CFU in the lungs of the *furA*/mc²114-infected mice were substantially fewer than in the mc²114-infected mice and slightly more than in the mc²155-infected mice 7 days after intranasal infection of mice (Fig. 6D). Mouse lethality assays showed that 60% of the mice survived the

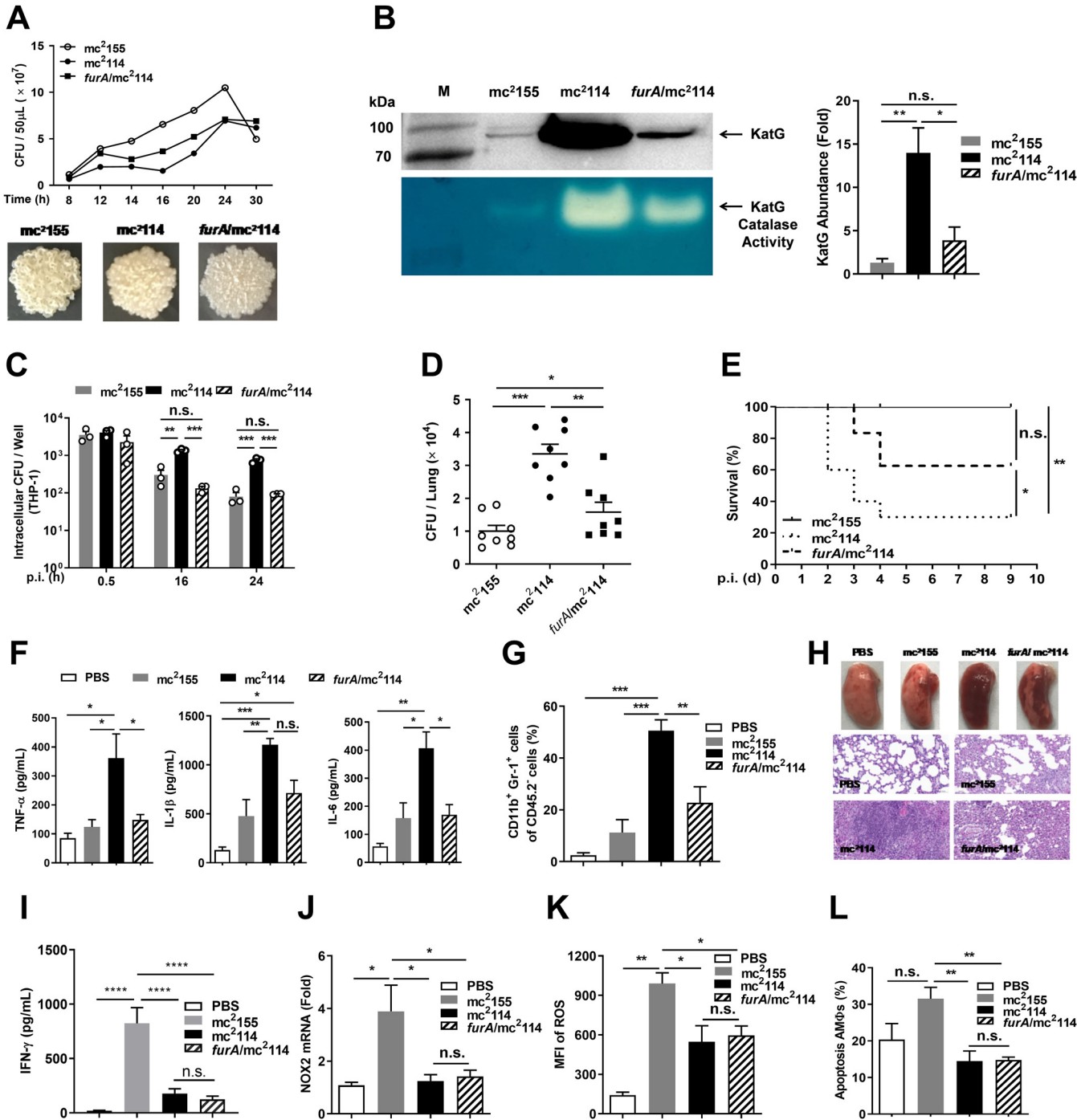

**FIG 6** Complementation of mc²114 with *furA* reversed inflammatory response and alleviated virulence in mice significantly. (A) Growth curve (top) and colonial morphology (bottom) of mc²155, mc²114, or *furA*/mc²114. (B) The expression and enzymatic activity of KatG of indicated *Msmeg* strains were measured as in Fig. 1D. Data are presented as means ± SEM from three independent experiments and representative images are shown. *, $P < 0.05$ and **, $P < 0.01$ by one-way ANOVA with Tukey's test. (C) Intracellular growth of mc²155, mc²114, or *furA*/mc²114 in THP-1 cells was determined at different time points after infection. (D) Mice were infected i.n. with *Msmeg* strains as indicated ($1 \times 10^7$/mouse). Seven days after infection, CFU in the lungs were determined. Data are presented as means ± SEM from two independent experiments and analyzed by two-tailed unpaired Mann-Whitney *t* tests. (E) Mice were infected i.v. with *Msmeg* strains ($3 \times 10^7$/mouse) as indicated ($n = 10$) and the survival rate was analyzed by the Gehan-Breslow-Wilcoxon test. Twenty-four hours after i.n. infection, (F) the concentrations of TNF-$\alpha$, IL-1$\beta$, IL-6, and (I) IFN-$\gamma$ in the lung homogenates were measured by ELISA, (G) the neutrophils (CD11b⁺ Gr-1⁺) in the lungs were analyzed by flow cytometry, (H) the lung sections were H&E stained and photographically recorded (middle and bottom; scale bar, 100 $\mu$m), and (J) the levels of NOX2 mRNA in lung cells were determined by qRT-PCR. AMΦs were isolated 24 h after infection for ROS detection (K), or at 6 h after infection for apoptosis measurement (L) by flow cytometry. Data are presented as means ± SEM from two independent experiments. *, $P < 0.05$; **, $P < 0.01$; ***, $P < 0.001$; ****, $P < 0.0001$; and n.s., not significant by one-way ANOVA with Tukey's test.

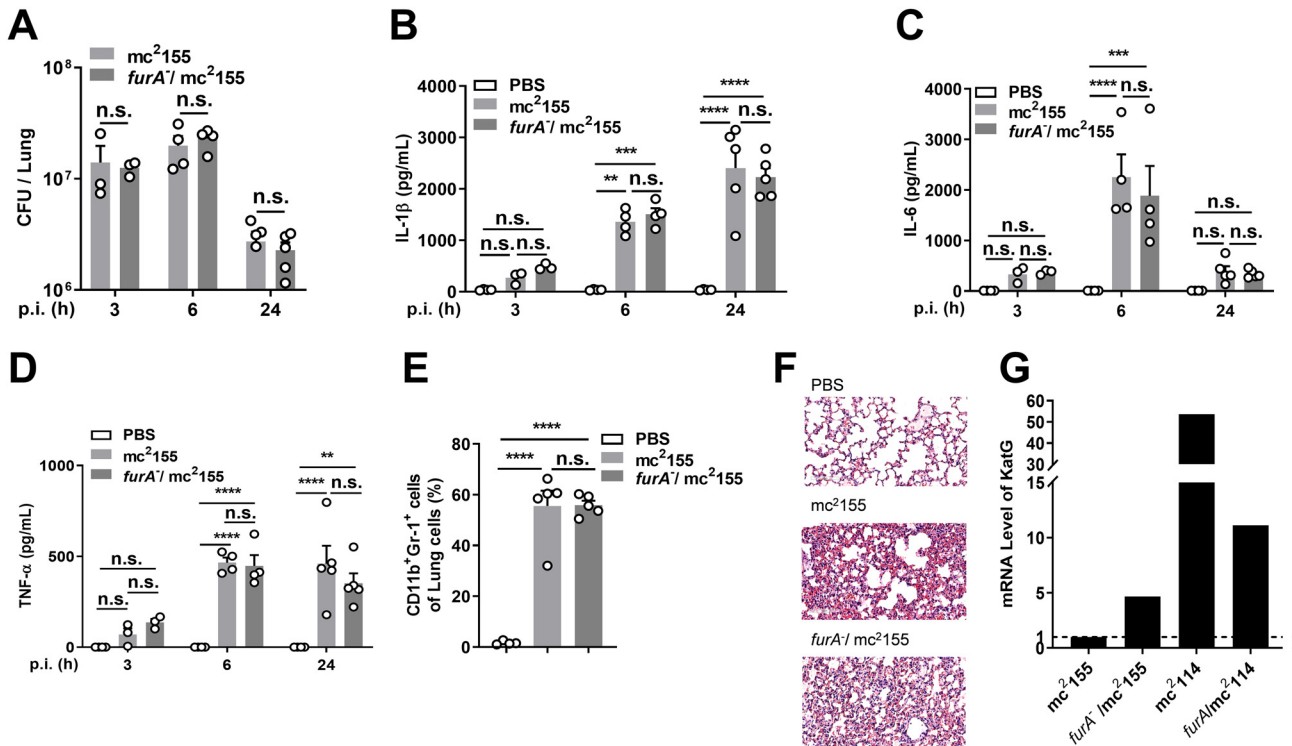

**FIG 7** FurA-deficient mc²155 (*FurA⁻*/mc²155) failed to induce higher levels of inflammatory responses in mice. Mice were infected i.n. with *Msmeg* strains (1 × 10⁷/mouse) as indicated. CFU in the lungs (A) and the concentrations of IL-1$\beta$ (B), IL-6 (C), and TNF-$\alpha$ (D) in the lung homogenates were determined at different time points after infection. Twenty-four hours after infection, (E) the neutrophils (CD11b⁺ Gr-1⁺) in the lungs were analyzed by flow cytometry, (F) the lung sections were H&E stained and photographically recorded (middle and bottom; scale bar, 100 $\mu$m). Data are presented as means ± SEM from two independent experiments. **, $P < 0.01$; ***, $P < 0.001$; ****, $P < 0.0001$; and n.s., not significant by one-way ANOVA with Tukey's test. (G) The expression of KatG of indicated *Msmeg* strains were examined by qRT-PCR.

*furA*/mc²114 infection 4 days after the intravenous challenge, and only 30% survived the mc²114 infection (Fig. 6E). In addition, the levels of inflammatory cytokines (TNF-$\alpha$, IL-1$\beta$, and IL-6) and neutrophils in the lungs of the *furA*/mc²114-infected mice reduced to levels similar to those of mc²155-infected mice (Fig. 6F and G). Lung histopathology examination of intranasally infected mice revealed that lung consolidation and hemorrhage were milder in *furA*/mc²114-infected than in mc²114-infected mice and more severe than in mc²155-infected mice (Fig. 6H), indicating that FurA contributes to mc²114-induced hyper-inflammatory response. However, responses of IFN-$\gamma$, NOX2, ROS, and AM$\Phi$ apoptosis were similar in *furA*/mc²114- and mc²114-infected mice (Fig. 6I–L), suggesting that other SNPs in mc²114 may play a role in these responses. In addition, *furA* complementation substantially reversed KatG expression but failed to rescue the ROS reduction in mice, suggesting that the ROS reduction in mc²114-infected mice is not attributed to KatG upregulation; it is rather attributed to the reduced NOX2 or compensatory mechanisms of other oxidative metabolite scavenging enzymes independent of FurA (16, 35).

**FurA-deficient mc²155 failed to induce severe inflammatory responses compared to mc²155 in mice.** To determine whether the role of FurA in inflammatory cytokine response involves other SNPs in mc²114, FurA-deficient mc²155 (*furA⁻*/mc²155) was generated (Fig. S3, Table S2), which contains the same amino acid substitution (A28V) as in *furA* in mc²114 but no other SNPs. The mice were intranasally infected, and CFU in the lungs were detected at indicated time points. We found that the number of CFU was similar in *furA⁻*/mc²155-infected mice and mc²155-infected mice (Fig. 7A). ELISA showed that the productions of IL-1$\beta$, IL-6, and TNF-$\alpha$ in the lungs of *furA⁻*/mc²155- and mc²155-infected mice were comparable (Fig. 7B–D), and the neutrophil counts in the lungs (Fig. 7E) and pathological changes in lung sections (Fig. 7F) of *furA⁻*/mc²155- and mc²155-infected mice were similar. KatG expression examined by qRT-PCR showed that the point mutation of *furA*

caused an increase in KatG (4.67-fold that of mc$^2$155) (Fig. 7G), which confirmed FurA deficiency in *furA*$^-$/mc$^2$155. However, it was lower than that in mc$^2$114 (53.67-fold that of mc$^2$155) with the same mutation of *furA*. These results indicate that FurA functions differently with and without the genes harboring the SNPs.

## DISCUSSION

Resistance to oxidative metabolites is a critical mechanism for *Mtb* survival within host phagocytes. In this study, a nonpathogenic *Msmeg* underwent genetic alterations and displayed slow growth, high H$_2$O$_2$ resistance, and longer survival in mice with the attenuated ROS response and hyper-inflammatory damage in the lungs after evolutionary adaption to H$_2$O$_2$. Whole-genome sequencing showed that one SNP is on *furA*, which resulted in increased KatG expression and hyper-inflammatory damage in the lungs. A wild-type *furA* complementation of mc$^2$114 partially rescued the slow growth and persistence in mice. The results indicate that *furA*, one of the genes harboring genetic changes, plays a dominant role in mc$^2$114 survival in hosts.

In *Mtb* infection, hosts show immune responses that result in the production of ROS and proinflammatory mediators (36), which is required to prevent *Mtb* dissemination. However, excessive inflammation causes lung parenchymal damage resulting in reduced disease tolerance and the onset of active disease (37). ROS play important roles in the host defense against mycobacterial infection and are central to the progression of inflammation (1, 38, 39). Studies have shown that proinflammatory cytokine responses to mycobacteria are mediated by TLR2-induced ROS generation (40, 41). KatG is a catalase/peroxidase that detoxifies ROS. FurA deficiency causes KatG overexpression in mc$^2$114, which can reduce ROS. We found that mc$^2$114 intensified inflammatory cytokine responses with ROS reduction in mice. However, *furA* complementation reversed the inflammation but not the ROS reduction, indicating that there is no direct causal relationship between ROS and inflammatory responses in mc$^2$114-infected mice. The hyper-inflammatory response may have also resulted from the persistence of mc$^2$114 in the lungs. We observed higher levels of inflammatory cytokines in the lungs in mc$^2$114-infected mice than in mc$^2$155-infected mice 6 h after infection, but the bacterial counts were similar in the mc$^2$114 and mc$^2$155 groups within 1 day after infection (Fig. 2A and 5A). This indicated that the hyper-inflammation was not caused by mc$^2$114 persistence, and FurA was involved in the regulation of the inflammatory response (42, 43). Our result indicated that the induced ROS production in the host was FurA-independent and further exploration was required.

There is a direct causal relationship between *Mtb* virulence and the ability of *Mtb* to inhibit macrophage apoptosis (29, 44). Apoptotic lung cells were increased in mc$^2$155-infected mice but not those infected by mc$^2$114, indicating that the restrained host apoptosis response is involved in the increased survival of mc$^2$114 in the lungs. ROS-mediated apoptosis is TNF-$\alpha$ dependent (29). We found reduced ROS and apoptosis but increased TNF-$\alpha$ production in mc$^2$114-infected mice. Similarly, in *furA*/mc$^2$114-infected mice, TNF-$\alpha$ response reversed but responses of ROS and apoptosis remained as those in mc$^2$114-infected mice. These findings suggest that apoptosis response to *mycobacteria* is also TNF-$\alpha$ independent.

Our study showed that mc$^2$114, which carries SNPs in *furA* and other genes, was pathogenic in mice, and *furA* complementation significantly suppressed its pathogenicity with other SNPs remaining. Moreover, hyper-inflammation was induced by mc$^2$114 but not *furA*$^-$/mc$^2$155. These observations indicate that FurA functions differently with the genetic backgrounds of mc$^2$114 and mc$^2$155 and suggest that FurA-regulated inflammatory response requires the genes harboring SNPs in mc$^2$114 that are not in mc$^2$155. Further study should determine whether some of the SNP-harboring genes individually or collaboratively become responsive to FurA in mc$^2$114 and whether they are present in *Mtb*.

The slow growth is a characteristic feature of pathogenic mycobacteria, such as *Mtb* and *Mycobacterium leprae*, relative to nonpathogenic mycobacterial species, suggesting a relationship between mycobacterial growth rate and virulence (45). Slow-growing mycobacteria enter macrophages more efficiently and induce lower levels of ROS in the cells than fast-growing mycobacteria (46). We found that mc$^2$114 had a lower growth rate than its parental strain

mc²155. The better survival in AMΦs associated with slow growth was also found in the $H_2O_2$-adapted *Msmeg* strain mc²51 (3, 4). In addition, mc²114 altered colony morphology, suggesting that the cell wall components were modified to adapt to $H_2O_2$. The unique cell wall components of *Mtb* act as potent scavengers of oxygen radicals and are also instrumental in determining the pattern of inflammatory responses (47, 48). Further studies should determine whether the lower growth rate and colony morphology alteration contribute to the exacerbated inflammatory response and resistance to ROS and whether FurA affects inflammatory response through the regulation of cell wall synthesis.

Collectively, the results demonstrate that mechanisms involved in mycobacterial pathogenesis are evolved through the adaptation of *Msmeg* to $H_2O_2$. The $H_2O_2$-adapted *Mycobacterium* showed better survival in mice with restricted bacterial elimination and augmented inflammatory lung tissue damage. The findings demonstrate that *furA* plays a critical role in the regulation of mycobacterial pathogenesis and suggest that mc²114 with $H_2O_2$ adaptation-related genetic changes can be used as a mycobacterial model for studying virulent factors and a potential *Mtb* vaccine vector.

## MATERIALS AND METHODS

This study was performed in strict adherence to the recommendations of the Guide for the Care and Use of Laboratory Animals of the Institute of Microbiology, Chinese Academy of Sciences (IMCAS) Ethics Committee. The protocol was approved by the Committee on the Ethics of Animal Experiments of IMCAS (permit number: APIMCAS 2015010). The mice were bred under specific pathogen-free conditions in the laboratory animal facility at IMCAS. All animal experiments were conducted under isoflurane anesthesia, and all efforts were made to minimize suffering.

**Bacteria and growth condition.** The $H_2O_2$-resistant *Msmeg* strain mc²114 was generated using adaptive evolutionary selection with $H_2O_2$ as described previously (10). mc²114, its parental strain (mc²155), and the *furA*/mc²114 strain were cultured as previously described (49). The CFU of *Msmeg* were detected by plating serial dilutions of cultures onto Middlebrook 7H10 agar plates followed by incubation at 37°C in an atmosphere of 5% $CO_2$ for 3 days.

**KatG activity assay, immunoblotting, and INH susceptibility testing.** Crude extracts of each *Msmeg* strain were prepared (50), and the catalase activity of KatG was assessed using the method described by Wayne and Diaz (51). Briefly, 20 $\mu$g of the crude extracts were detected in native 7.4% (wt/vol) polyacrylamide gels. After electrophoresis, the gels were washed and soaked in distilled water containing 5 mM $H_2O_2$ for 10 min. After washing, the gels were incubated with a solution containing 2% (wt/vol) ferric chloride and 2% (wt/vol) potassium ferricyanide until they began to turn deep green. After washing, the protein bands with catalase activity were detected as achromatic bands in the deep green background.

Immunoblotting was carried out after the SDS-PAGE and transfer of proteins to polyvinylidene difluoride (PVDF) membranes (Millipore immobilon-P, Billerica, MA, USA), followed by incubation with an anti-KatG (diluted 1:2,000) polyclonal antibody. After washing with TBS with 0.05% Tween 20 (TBST), the blots were incubated with goat antimouse IgG conjugated to horseradish peroxidase (HRP) at a dilution of 1:10,000. The bands of KatG were visualized by chemiluminescence using an enhanced chemiluminescence (ECL) substrate (Thermo Fisher Scientific, catalog number 32106) and quantified with Tanontanon 5200 (Tanon 5200, Shanghai, China). The MIC for INH was determined by inoculating dilutions of an 18-h 7H9-ADS culture (50 $\mu$L) into a 96-well cell culture plate containing 50 $\mu$L of fresh medium per well. Each well was supplemented with one of eight concentrations of INH ranging from 0.3125 to 40 $\mu$g/mL, and the plates were read after 3 days to ascertain the MIC.

**Mice and bacterial infection.** Female-specific pathogen-free C57BL/6 mice (6 to 8 weeks) were purchased from Vital River (Beijing). For intranasal infection with *Msmeg*, the mice were anesthetized with an intraperitoneal injection of pentobarbital sodium (60 mg/kg);1 to 2 × $10^7$ CFU/50 $\mu$L of PBS of mc²114 or mc²155 were administered drop by drop through the left nostril of each mouse. The bacterial load throughout the infection was monitored by harvesting lungs at the indicated times after the mice were euthanized and serial dilutions on 7H10 were poured in the agar plates. For survival assays, the mice were intravenously infected with 3 × $10^7$ CFU of *Msmeg* (mc²114 or mc²155) per animal in 100 $\mu$L PBS. The survival of the mice was monitored daily. The dose of infection was confirmed on day one postinfection by plating whole lung homogenates from three mice on 7H10 agar.

**Lung sample preparation.** Single-cell suspension of lung tissue was centrifuged, and the supernatant was collected for measuring cytokines by ELISA. The cells were assessed by flow cytometry analysis and qRT-PCR. Some of the samples of lung tissues were fixed in 4% (vol) formalin and embedded in paraffin, and then sections were cut and stained with hematoxylin and eosin (H&E) for evaluation of tissue pathological changes. The BALF was obtained by flushing the lungs three times with 0.5 mL of PBS supplemented with 0.2% (wt/vol) bovine serum albumin (BSA) (52) and stored on ice until further processing.

**Isolation of alveolar macrophages.** BALF samples from mice were obtained as described above. After centrifugation at 1,600 rpm for 5 min at 4°C, cell pellets were seeded in a 24-well plate (Corning, USA) and incubated for 12 h at 37°C 5% $CO_2$ in DMEM medium containing 10% fetal bovine serum (FBS). Nonadherent cells were removed by rinsing the wells with PBS, and the adhered cells were detached by adding 0.25% Trypsin/EDTA (Gibco) after washing three times with 1× PBS. To determine the purity of the adhered cells

(AMΦs), cells were stained with antibodies directed to F4/80 (clone: BM8, eBioscience, USA) and CD11b (clone: M1/70, Biolegend, USA) and analyzed by flow cytometry. The purity of AMΦs was ≥ 95%.

**AMΦ intracellular CFU counting.** For intracellular CFU counting *in vivo*, AMΦs isolated from the BALF of infected mice were washed and incubated with RPMI 1640 medium containing 10% FBS and supplemented with 15 $\mu$g/mL gentamicin to kill extracellular bacteria at 37°C for 2 h. The cells were subsequently washed, trypsinized, and lysed in sterile, cold PBST (PBS with 0.05% Tween 20) for 30 min. The lysates were vortexed, diluted, and poured onto 7H10 agar plates.

**Intracellular CFU counting *in vitro*.** THP-1 cells were seeded in a 24-well plate ($2 \times 10^5$ cells/well) and cultured in RPMI 1640 medium with 10% FBS, overnight at 37°C under 5% $CO_2$. The cells were treated with 100 nM Phorbol-12-myristate-13-acetate (PMA) (Sigma) during the overnight incubation for differentiating into macrophage-like cells. The cells were infected with *Msmeg* at a multiplicity of infection (MOI) of 10. At different time points after infection, the extracellular bacteria were killed with gentamicin, and the intracellular CFU in THP-1 cells were countered as in AMΦ assays above.

**ROS measurement.** The ROS of AMΦs from BALF were assessed with a reactive oxygen species assay kit (Beyotime, Shanghai, China). The BALF samples were centrifuged at 1,600 rpm for 5 min at 4°C, and the cell pellets were resuspended in 1 mL serum-free RPMI 1640 with 2,7-dichlorofluorescein diacetate (1:2,000) and stained with specific markers for AMΦs (see above) for 25 min at 37°C. The ROS were assessed with flow cytometry based on the mean fluorescence intensity of dichlorofluorescein in the cells, which reflects the intracellular ROS levels. Flow cytometry was performed using a FACSCanto flow cytometer and further analyzed with FlowJo software (TreeStar, Ashland, OR, USA).

**Assessment of apoptosis.** The apoptosis of AMΦs and THP-1 cells were assessed with the fluorescein isothiocyanate annexin V/PI apoptosis detection kit (BioLegend, USA) according to the manufacturer's instructions. The cell pellets were resuspended in 100 $\mu$L annexin binding buffer with fluorescein isothiocyanate-conjugated annexin V (1:500) and stained for 15 min in a dark at room temperature. Immediately before collection, 400 $\mu$L of annexin binding buffer and 40 $\mu$L (100 $\mu$g/mL) of propidium iodide (PI) were added to each sample. The percentage of apoptotic macrophages (Annexin V$^+$/PI$^-$) was determined by flow cytometry.

**Western blot analysis of apoptotic proteins.** Western blotting was performed to determine the mechanisms of apoptosis. THP-1 cells ($3 \times 10^5$/well) were infected with mc$^2$155 or mc$^2$114 (MOI = 10) at 12 h pi. Cells were washed and then lysed with ice-cold radioimmunoprecipitation assay (RIPA) buffer (Beyotime, Shanghai, China) containing protease inhibitors for 30 min at 4°C. Total cell lysates were analyzed for protein contents using bicinchoninic acid (BCA) assay (Tiangen, Beijing, China). Samples (20 $\mu$g per well) were electrophoresed on 12% SDS-PAGE gels and transferred to PVDF membranes at 150 mA for 1 h, blocked in 5% nonfat dry milk in TBST for 2 h at room temperature and subsequently incubated overnight at 4°C with anti-Bax (ab32503) and anti-Bcl-2 (ab32124) (Abcam, Cambridge, MA, USA), respectively. Anti-$\beta$-actin antibody (TransGen, Beijing, China) was used as an internal control. All antibodies were used at a dilution of 1:2,000 in TBST and 5% (wt/vol) BSA. Following three washes with TBST, the blots were incubated with goat antirabbit IgG conjugated to HRP (ZSGB-Bio, Beijing, China) at a dilution of 1:10,000 in blocking buffer for 45 min at room temperature. The blots were developed with ECL substrate (Thermo Fisher Scientific, catalog number 32106) after washing, and exposed to autoradiography film (Carestream Rochester, NY, USA). Images of bands were scanned using Photoshop software (Adobe Systems, San Jose, CA, USA).

**RNA isolation and qRT-PCR.** The total RNA of the lungs was isolated using TRIzol reagent (Invitrogen, Carlsbad, CA, USA). Reverse transcription was performed using high-capacity M-MLV reverse transcriptase (Applied Biosystems, Life Technologies, CA, USA), as recommended by the manufacturer. Transcript products were amplified with SYBR *Premix Ex Taq* II (TaKaRa, Dalian, China) on a Roche 480 II (Roche) using the following specific primer sets (5'–3'): NOX2 forward: CCAGTGAAGATGTGTTCAGCT, NOX2 reverse: GCACAGCCA GTAGAAGTAGAT; GAPDH forward: CATGGCCTTCCGTGTTCCTA, GAPDH reverse: GCGGCACGTCAGATCCA.

The levels of relative gene expression were evaluated using the $2^{-\Delta\Delta Ct}$ method and were normalized to *Gapdh* mRNA. The relative expression of these genes in PBS-treated controls was normalized to 1.0.

**Isolation of pulmonary mononuclear cells and flow cytometry analysis.** Mouse pulmonary mononuclear cells were isolated as described by Gibbings and Jakubzick (53) with modifications. Briefly, the lungs were harvested after perfusion with PBS, cut into small pieces, and digested with collagenase type IV (540 U/mL; Worthington Biochemical Corporation, Lakewood, NJ, USA) in RPMI1640 medium for 45 min at 37°C. Single-cell suspensions were prepared by mechanical dissociation of lung tissue through a 70-$\mu$m nylon mesh; cells were harvested, suspended in PBS, and isolated using standard density gradient techniques. The isolated pulmonary mononuclear cells were stained for a surface marker with anti-CD3 PerCP (clone: 145-2C11, Biolegend, CA, USA) and anti-NK1.1 PE (clone: PK136, Biolegend, USA) for NKT cells; with anti-CD3 PerCP and anti-$\gamma\delta$ TCR FITC (clone: eBioGL3, eBioscience, CA, USA) for $\gamma\delta$ T cells; and with anti-CD11b FITC and anti-Gr-1 APC (clone: RB6-8C5; Biolegend, CA, USA) for neutrophils. For intracellular staining, fixed cells were permeabilized and stained with anti-IFN-$\gamma$ APC (clone: XMG1.2; eBioscience, CA, USA) for 60 min in the dark at 4°C. Cells were washed with 1 mL of permeabilization buffer and resuspended in staining buffer for analysis. Samples were analyzed using a FACSCalibur or FACSCanto flow cytometer (BD Biosciences NJ, USA) and the FlowJo software (Treestar, Ashland, OR, USA).

**Determination of cytokines by ELISA.** The concentrations of TNF-$\alpha$, IL-1$\beta$, IL-6, IFN-$\gamma$, and IL-12p70 in supernatants of lung tissue homogenate were determined by ELISA (eBioscience, CA, USA) according to the manufacturer's instructions.

**Histological analysis of lung tissue.** The lungs were removed and fixed with 4% paraformaldehyde. After embedding the tissue in paraffin, sections were stained with hematoxylin-eosin according to the standard protocol and examined using a Leica SCN400 device (Wetzlar, Germany).

**Generation of *furA*/mc²114 and *furA⁻*/mc²155 strains.** For *furA*/mc²114, the coding region of *furA* was amplified from *Msmeg* genomic DNA using the following PCR conditions: 98°C for 3 min, 32 cycles of 98°C for 30 s, 60°C for 30 s, and 72°C for 30 s, and 72°C for 6 min. The following specific primer set (5′–3′) was used for amplifying the *furA* gene: pMV361-MSM3460F: ACGTGAATTCCCCTCTCGGGCGGAGTTC; pMV361-MSM3460R: ACGTAAGCTTTCAAGGGCCGTCCACCGAG. The PCR fragment was cloned into the integrating vector pMV361 (54) to yield pMV361-*furA*. The constructed plasmid was transformed into *Msmeg* strain mc²114 via electroporation.

The *furA⁻*/mc²155 strain carrying a point mutated site on *furA* was created following the previously published method (3). Briefly, the single-strand DNA (155-*furA*–Lag) was used to direct the site mutagenesis of *furA* using a recombination system based on phage Che9c gp61 recombination proteins (55). The 155-*furA*–Lag ordered from Sangon Biotech Company (Shanghai, China) (Table S2) is used for the allelic exchange of *furA* locus in mc²155 targeting the lagging strand of *furA* on the genome and Hyg-Lag (Table S2) for reversing hygromycin susceptibility targeting the *hyg* gene on the plasmid pJV62. mc²155 harboring plasmids pJV62 (pJV62/mc²155) were grown in 7H9 with 0.2% succinate at an optical density at 600 nm (OD$_{600}$) of 0.02 and incubated overnight to OD$_{600}$ of 0.5, then acetamide (0.2%) was added. After 3-h incubation, the competent pJV62/mc²155 cells were electroporated with 100 ng of 155-*furA*–Lag and 100 ng of Hyg-Lag harboring DNA fragments of *furA⁻* and *hyg*, respectively. The electroporated pJV62/mc²155 cells were recovered in 7H9 for 4 h with shaking at 37°C. Then the recovered cells were plated on 7H10 agar supplemented with 50 mg/L hygromycin. MAMA-PCR assay was used for the detection and selection of *furA⁻*/mc²155. The primers (*furA*-F, 155-*furA⁻*-F, *furA*-R) were designed to distinguish between wild type *furA* and mutant *furA* according to the previous study (56) (Table S2). The PCR fragments carrying a mutated site in *furA* (*furA⁻*) were sequenced to confirm by Genewiz Company (Suzhou, China).

**Statistical analysis.** Each experiment was performed at least twice with 3 to 6 mice or samples per group. CFU were analyzed by two-tailed unpaired Mann-Whitney $t$ tests, survival was analyzed using the log-rank test, and others were analyzed by one-way ANOVA followed by Tukey's multiple-comparison test using GraphPad Prism software (version 7.0 for Windows; GraphPad Software, San Diego, CA, USA). Significance was reached at $P < 0.05$. Data were expressed as the means $\pm$ SE of the mean (SEM).

**Data availability.** We declare that all data supporting the findings of this study are available within the article and its supplementary information files or from the corresponding author upon reasonable request. The genome information was deposited in China National Microbiology Data Center (https://nmdc.cn/) with accession number NMDC40026043. The source data underlying Fig. 2 to 7 are provided as a Source Data file.

## SUPPLEMENTAL MATERIAL

Supplemental material is available online only.

**SUPPLEMENTAL FILE 1**, PDF file, 0.3 MB.

## ACKNOWLEDGMENTS

We thank Hao J (Institute of Biophysics, Chinese Academy of Sciences, Beijing, China) for help with the histological examination; Zhou Y (Institute of Microbiology, Chinese Academy of Sciences, Beijing, China) for advice on flow cytometry; and Xu M (University of Chinese Academy of Sciences, Beijing, China) for immunoblot analysis. This research was supported by funding from the National Natural Science Foundation of China grant 31872743 (to B.W.), grant 31970136 (to K.M.), grant 31600737 for Young Scholars (to X. F), and International Joint Research Project of the Institute of Medical Science, University of Tokyo Extension-2019-K3006 (to K.M.).

B.W. conceived the original idea, designed the experiments, and wrote the manuscript. K.M. designed the adaptation experiment and genetic analysis and provided suggestions for writing the manuscript. X.F. and B.Z. performed most of the experiments and analyzed the data. W.Z. generated *furA⁻*/mc²155. N.L. provided advice on and contributed to some of the experiments.

We declare no competing financial interests.

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
