## [Reviewer comments · Microbiology Spectrum]

Microbiology Spectrum

Co-evolution of *furA*-regulated hyper-inflammation and mycobacterial resistance to oxidative killing through adaptation to hydrogen peroxide

Xin Fan, Bei Zhao, Weishan Zhang, Ning Li, Kaixia Mi, and Beinan Wang

Corresponding Author(s): Beinan Wang, Institute of Microbiology Chinese Academy of Sciences

Review Timeline:

Submission Date:	December 31, 2022
Editorial Decision:	January 27, 2023
Revision Received:	March 29, 2023
Editorial Decision:	April 27, 2023
Revision Received:	May 24, 2023
Accepted:	May 25, 2023

Editor: Amit Singh

Reviewer(s): The reviewers have opted to remain anonymous.

Transaction Report:

DOI: <https://doi.org/10.1128/spectrum.05367-22>

January 27, 2023

Prof. Beinan Wang
Institute of Microbiology, Chinese Academy of Sciences
Key Laboratory of Pathogenic Microbiology and Immunology
1 Beichen West Road, Chaoyang District
Beijing 100101
China

Re: Spectrum05367-22 (**Co-evolution of *furA*-regulated hyper-inflammation and mycobacterial resistance to oxidative killing through adaptation to hydrogen peroxide**)

Dear Prof. Beinan Wang:

Both the reviewers recognized the importance of the manuscript. However, several concerns were raised related to CFU numbers, viability of macrophages, and statistical significance. Please address these issues constructively.

Link Not Available

Sincerely,

Amit Singh

Journals Department
Reviewer comments:

Reviewer #1 (Comments for the Author):

In this manuscript authors have generated a H₂O₂-resistant strain (mc2 114) via evolutionary H₂O₂ adaptation in vitro. By performing macrophage and mice infections they showed that the mc2 114 was persistent in the lungs and caused high lethality in mice with restricted responses of NOX2, ROS, IFN- γ , decreased macrophage apoptosis, and over-expressed inflammatory cytokines in the lungs. One of the 29 SNPs identified by whole genome sequencing was in *furA* gene that caused overexpression of KatG. My comments are as follows:

1. Line 86-87 "However, Msmeg is much less ROS resistant". Is this true? A previous study (Free Radic Biol Med. 2015;84:344-354. doi:10.1016/j.freeradbiomed.2015.03.008) has shown that "Mtb had diminished capacity to restore cytoplasmic redox balance in comparison with Mycobacterium smegmatis (Msm), a fast growing nonpathogenic mycobacterial species". Authors need to clarify the above statement with appropriate references.
2. Fig 2A, was the initial bacteria load for Day 0 same for the 2 strains? Looks like 1-day post-infection itself the numbers are different for the 2 strains.
3. Fig 2B, I am not convinced why the difference at 6 and 24 hrs is significant but not at 120 hrs pi? Please explain.
4. Fig 2C, authors need to perform viability assay for THP1 cells (eg MTT) to make sure that the survival difference is not because of differential viability of THP1 cells between the 2 bacterial strains.
5. For Fig 2D, authors infected mice intravenously while for Fig 2A mice were intranasally infected. Why? What was the bacterial load when infection was done intravenously?
6. Statistical comparison in Fig 3A (and in several other figures) is confusing. Authors are showing asterisk above the bar as well as comparing 2 bars (155 vs 114) with a line, and then putting "ns" sign above 114. Please simplify this and explain properly which conditions are being compared? Please correct it for other figures as well.

Reviewer #2 (Comments for the Author):

The manuscript titled "Co-evolution of furA-regulated hyper-inflammation and mycobacterial resistance to oxidative killing through adaptation to hydrogen peroxide" by Fan et al., describes the generation of an H₂O₂-resistant mutant of *M. smegmatis* (named as mc2114) with multiple mutations through an adaptive evolutionary screen. The Msemg mc2114 mediated severe inflammatory lung injury and higher lethality in mice associated with overexpression of inflammatory cytokines. Further data suggest that FurA-regulates pulmonary inflammation in mice along with down-regulation of NOX2, ROS, IFN- γ responses, and macrophage apoptosis. This is a fascinating and well-written manuscript with carefully designed experiments. I have the following comments/suggestions for the authors.

1. Figure 1D and 6B. Please clarify if the data are from one experiment only, as there are no error bars/statistical analysis.
2. Figure 2C, data shall be presented as CFU/million THP-1 or CFU/104 THP-1 cells. CFU/well of THP-1 does not represent the correct way of representation as it could lead to misinterpretation of data.
3. Mycobacterium is a genus and hence shall be italicized in lines 84 and 28.
4. The sentence in lines 138-141 lacks clarity. Please modify.
5. Line 149-151, please mention the infection dosage to clarify the text. Although the dosage is mentioned in the figure legend, bringing it here as well will make it aptly clear.
6. Lines 166-168, is it possible to analyze the levels of NADPH or its activity?
7. Line 220-222, the authors have made a clean mutant, as claimed in the supplementary data. But mentioning ectopic expression confuses the readers. Please clarify.
8. Discussion section. Data presented in the manuscript suggest a correlation between the capability of mycobacterial strains to cause apoptosis and the protection of the host. The discussion section may be improved further but discussing these findings.

Staff Comments:

Preparing Revision Guidelines

For complete guidelines on revision requirements, please see the journal Submission and Review Process requirements at

<https://journals.asm.org/journal/Spectrum/submission-review-process>. **Submissions of a paper that does not conform to Microbiology Spectrum guidelines will delay acceptance of your manuscript. "**

Please return the manuscript within 60 days; if you cannot complete the modification within this time period, please contact me. If you do not wish to modify the manuscript and prefer to submit it to another journal, please notify me of your decision immediately so that the manuscript may be formally withdrawn from consideration by Microbiology Spectrum.

Response to reviewers

Response to reviewer #1:

In this manuscript authors have generated a H₂O₂-resistant strain (mc² 114) via evolutionary H₂O₂ adaptation in vitro. By performing macrophage and mice infections they showed that the mc²114 was persistent in the lungs and caused high lethality in mice with restricted responses of NOX2, ROS, IFN- γ , decreased macrophage apoptosis, and over-expressed inflammatory cytokines in the lungs. One of the 29 SNPs identified by whole genome sequencing was in *furA* gene that caused overexpression of KatG. My comments are as follows:

1. Line 86-87 "However, *Msmeg* is much less ROS resistant". Is this true? A previous study (Free Radic Biol Med. 2015;84:344-354. doi:10.1016/j.freeradbiomed.2015.03.008) has shown that "*Mtb* had diminished capacity to restore cytoplasmic redox balance in comparison with *Mycobacterium smegmatis* (*Msm*), a fast growing nonpathogenic mycobacterial species". Authors need to clarify the above statement with appropriate references.

We thank the reviewer for bringing out the issue. The report studies *Mtb*'s response to endogenous ROS generated within *Mtb* and shows that *Mtb* has a diminished capacity to restore intra-mycobacterial redox balance in comparison with *Msmeg*, leading to *Mtb*, but no *Msmeg*, extremely susceptible to inhibition by compound-induced endogenous ROS, raising a possibility of targeting intra-mycobacterial redox metabolism for controlling TB infection. The study is very interesting and supportive for our findings. *Mtb* is known to secrete large amounts of antioxidant enzymes such as catalase and superoxide dismutase to dissipate host-generated oxidative stress for persistence in host cells, indicating differences of *Mtb* in defense mechanisms to tolerate ROS produced inside or outside the bacterial cells. In contrast, *Msmeg* contains nonsecretory forms of antioxidant enzymes and is tolerant to endogenous ROS but more sensitive to killing by exogenous ROS. Our results indicate that mc²114 evolves the ability to dissipate exogenous ROS in the evolutionary adaptation to H₂O₂.

We add lines as "Although *Mtb* has a diminished capacity to restore endogenous

redox balance in comparison with *Msmeg* (1) it tolerates exogenous ROS and can persist in phagocytes. Whereas *Msmeg* is cleared from the lungs promptly, suggesting that *Msmeg* is less resistant to exogenous ROS (2), probably activities of these genes in *Msmeg* differ markedly (page 4, lines 86-90).

2. Fig 2A, was the initial bacteria load for Day 0 same for the 2 strains? Looks like 1-day post-infection itself the numbers are different for the 2 strains.

Mice were inoculated 2.44×10^7 CFUs of mc²155 or 1.51×10^7 CFUs of mc²114/mouse on day 0 (Fig. 2A). These inoculation doses were added in the figure legend of Fig. 2 in revised manuscript (page 33, lines 805-806).

3. Fig 2B, I am not convinced why the difference at 6 and 24 hrs is significant but not at 120 hrs pi? Please explain.

The CFUs of mc²114 were higher than that of mc²155 but not statistically significant because of variation in CFU numbers (We add *P*-value in the revised Fig. 2B).

4. Fig 2C, authors need to perform viability assay for THP1 cells (eg MTT) to make sure that the survival difference is not because of differential viability of THP1 cells between the 2 bacterial strains.

Viability assay was performed with a CCK-8 Assay Kit (KGA317, KeyGEN BioTECH, China) according the manufacturer's instructions and showed that following PMA stimulation, proliferation of THP-1 cells in non-infected group was lower than that in infected group, indicating that infection with *Msmeg* promotes growth of THP-1. When the two infected groups were compared, we found that proliferation of THP-1 was significantly higher in mc²155 infected than in mc²114 infected group at all examined time points (Fig. 1), ruling out that higher numbers of intracellular mc²114 is due to increased THP-1 cells and suggesting that mc²114 inhibits phagocyte proliferation and phagocytic killing, thus makes mc²114 survived better in macrophages (page 7, lines 153-156).

Fig. 1 Proliferation of THP-1 cells following mc²155 and mc²114 infection.

5. For Fig 2D, authors infected mice intravenously while for Fig 2A mice were intranasally infected. Why? What was the bacterial load when infection was done intravenously?

Intranasal model uses a lower dose ($1-2 \times 10^7$ CFUs / mouse) and determines abilities of the bacteria to colonize the lung. A higher dose (3×10^7 CFUs) was used in intravenous model to simulate systemic infection and evaluate ability of bacterial dissemination. The dosages were determined by preliminary experiments. We add the statement in the revised manuscript for better understanding (page 7, lines 157-159.)

6. Statistical comparison in Fig 3A (and in several other figures) is confusing. Authors are showing asterisk above the bar as well as comparing 2 bars (155 vs 114) with a line, and then putting "ns" sign above 114. Please simplify this and explain properly which conditions are being compared? Please correct it for other figures as well.

We thank the reviewer for the comment. All the comparisons in figures are changed using lines for clarity and consistency.

Response to reviewer #2:

The manuscript titled "Co-evolution of *furA*-regulated hyper-inflammation and mycobacterial resistance to oxidative killing through adaptation to hydrogen peroxide" by Fan et al., describes the generation of an H₂O₂-resistant mutant of *M. smegmatis* (named as mc²114) with multiple mutations through an adaptive

evolutionary screen. The *Msemg* mc²114 mediated severe inflammatory lung injury and higher lethality in mice associated with overexpression of inflammatory cytokines. Further data suggest that FurA-regulates pulmonary inflammation in mice along with down-regulation of NOX2, ROS, IFN- γ responses, and macrophage apoptosis. This is a fascinating and well-written manuscript with carefully designed experiments. I have the following comments/suggestions for the authors.

1. Figure 1D and 6B. Please clarify if the data are from one experiment only, as there are no error bars/statistical analysis.

Experiments in figures 1D and 6B were performed at least twice. One of the representing images was shown. Each image was scanned three times to get error bars/statistical analysis as shown in the revised manuscript (page 33, lines 801-803; page 35, lines 855-858).

2. Figure 2C, data shall be presented as CFU/million THP-1 or CFU/10⁴ THP-1 cells. CFU/well of THP-1 does not represent the correct way of representation as it could lead to misinterpretation of data.

Thank you for the correction. The Y-axis labeling is changed as suggested.

3. *Mycobacterium* is a genus and hence shall be italicized in lines 84 and 28.

Thanks for the correction. They are italicized in the revised manuscript (page 2, lines 28 and page 4, lines 84).

4. The sentence in lines 138-141 lacks clarity. Please modify.

We agree with the reviewer and modified the statement for clarity in the revised manuscript (page 6, lines 140-145).

5. Line 149-151, please mention the infection dosage to clarify the text. Although the dosage is mentioned in the figure legend, bringing it here as well will make it aptly clear.

Thanks for the suggestion. The infection dosage is included in the text (page 6, lines 138-140).

6. Lines 166-168, is it possible to analyze the levels of NADPH or its activity?

We thank the reviewer for this suggestion and measured NADPH activity. The results showed that NADP⁺/NADPH ratios were similar in mc²114 and mc²155

infected mice though higher levels of NADPH and NADP⁺ were in mc²114 infected than in mc²155 infected group. More statement and a new figure (Fig. S1) of the results are included in the revised manuscript (page 8, lines 179-184).

7. Line 220-222, the authors have made a clean mutant, as claimed in the supplementary data. But mentioning ectopic expression confuses the readers. Please clarify.

Thanks. We changed the sentence to avoid confusion (page 10, lines 232-234).

8. Discussion section. Data presented in the manuscript suggest a correlation between the capability of mycobacterial strains to cause apoptosis and the protection of the host. The discussion section may be improved further but discussing these findings.

We appreciate the reviewer's suggestion very much. A paragraph discussing mc²114 induced restriction of apoptosis is added in the revised manuscript (page 13, lines 309-317).

References

1. Tyagi P, Dharmaraja AT, Bhaskar A, Chakrapani H, Singh A. Mycobacterium tuberculosis has diminished capacity to counteract redox stress induced by elevated levels of endogenous superoxide. *Free Radic Biol Med.* 2015 Jul; 84: 344-354.
2. Jiang Z, Zhuang Z, Mi K. Experimental Evolution Reveals Redox State Modulates Mycobacterial Pathogenicity. *Front Genet.* 2022 Mar 16; 13:758304.

April 27, 2023

Prof. Beinan Wang
Institute of Microbiology Chinese Academy of Sciences
Key Laboratory of Pathogenic Microbiology and Immunology
1 Beichen West Road, Chaoyang District
Beijing 100101
China

Re: Spectrum05367-22R1 (**Co-evolution of *furA*-regulated hyper-inflammation and mycobacterial resistance to oxidative killing through adaptation to hydrogen peroxide**)

Dear Prof. Beinan Wang:

Thank you submitting the revised manuscript. Reviewer# 2 still have some lingering concerns, which need to be addressed. Please address the comments as constructively as possible.

Link Not Available

Sincerely,

Amit Singh

Journals Department
Reviewer comments:

Reviewer #1 (Comments for the Author):

I am satisfied with author's response to my comments.

Reviewer #2 (Comments for the Author):

The manuscript titled "Co-evolution of furA-regulated hyper-inflammation and mycobacterial resistance to oxidative killing through adaptation to hydrogen peroxide" by Fan et al., has improved through the revision. However, I still have the following concerns.

1. In figure 1D and 6B, scanning the same image 3 times to get the statistics is not appropriate. Data from different biological replicates shall be quantified and image shown could be a representative image.

Staff Comments:

Preparing Revision Guidelines

Please return the manuscript within 60 days; if you cannot complete the modification within this time period, please contact me. If you do not wish to modify the manuscript and prefer to submit it to another journal, please notify me of your decision immediately so that the manuscript may be formally withdrawn from consideration by Microbiology Spectrum.

Response to Reviewer #2

Reviewer #2 (Comments for the Author):

The manuscript titled "Co-evolution of furA-regulated hyper-inflammation and mycobacterial resistance to oxidative killing through adaptation to hydrogen peroxide" by Fan et al., has improved through the revision. However, I still have the following concerns.

1. In figure 1D and 6B, scanning the same image 3 times to get the statistics is not appropriate. Data from different biological replicates shall be quantified and image shown could be a representative image.

We understand and accept the reviewer's comments. The *Msmeg* strains were cultured, then the expression and enzymatic activity of KatG of the strains were measured (Fig. 1 and 2). The data in figures 1D and 6B were from three independent experiments (one done previously and two newly repeated). The figure legends of Fig.1D and 6B have modified correspondingly (page 33, lines 800-802; page 35, lines 850-852).

Fig.2

May 25, 2023

Prof. Beinan Wang
Institute of Microbiology Chinese Academy of Sciences
Key Laboratory of Pathogenic Microbiology and Immunology
1 Beichen West Road, Chaoyang District
Beijing 100101
China

Re: Spectrum05367-22R2 (**Co-evolution of *furA*-regulated hyper-inflammation and mycobacterial resistance to oxidative killing through adaptation to hydrogen peroxide**)

Dear Prof. Beinan Wang:

Your manuscript has been accepted, and I am forwarding it to the ASM Journals Department for publication. You will be notified when your proofs are ready to be viewed.

Sincerely,

Amit Singh
Editor, Microbiology Spectrum
